# ‘Shape-Up’, a Modified Cognitive-Behavioural Community Programme for Weight Management: Real-World Evaluation as an Approach for Delivering Public Health Goals

**DOI:** 10.3390/nu13082807

**Published:** 2021-08-16

**Authors:** Amber Soni, Rebecca J Beeken, Laura McGowan, Victoria Lawson, Paul Chadwick, Helen Croker

**Affiliations:** 1Research Department of Behavioural Science and Health, University College London, London WC1E 6BT, UK; amber.soni.17@alumni.ucl.ac.uk; 2Leeds Institute of Health Sciences, University of Leeds, Leeds LS2 9JT, UK; r.beeken@leeds.ac.uk; 3Centre for Public Health, School of Medicine, Dentistry and Biomedical Science, Queen’s University Belfast, Belfast BT9 7BL, UK; Laura.McGowan@qub.ac.uk; 4Institute for Global Food Security, School of Biological Sciences, Queen’s University Belfast, Belfast BT9 5DL, UK; 5Talking Therapies Southwark, Maudsley Hospital, London SE5 8AZ, UK; victoria.lawson@slam.nhs.uk; 6Centre for Behaviour Change, University College London, London WC1E 6BT, UK; p.chadwick@ucl.ac.uk; 7Population, Policy and Practice Research and Teaching Department, UCL Great Ormond Street Institute of Child Health, University College London, London WC1N 1EH, UK

**Keywords:** nutrition, physical activity, public health, obesity, weight management, behavioural programme, peer-learning, service evaluation, RE-AIM framework

## Abstract

Obesity is widespread, with serious health consequences; addressing it requires considerable effort at a public health level, incorporating prevention and management along with policies to support implementation. Behavioural weight-management programmes are widely used by public health bodies to address overweight and obesity. Shape-Up is an evidence-based programme combining a structured behavioural intervention (targeting nutrition and physical activity behaviours) within a peer-learning framework. This study was a service-evaluation of Shape-Up, as delivered in Rotherham by a local leisure provider, and included a secondary analysis of data collected in the community by service providers. The RE-AIM (Reach Effectiveness Adoption Implementation Maintenance) framework was used to explore programme effectiveness, implementation, and whom it reached. A total of 141 participants were included. Compared to local demographics, participants were older, at 48.9 (SD 14.47) years, with a lower employment rate (41% employed) and greater proportion female (67% female). Mean BMI was 38.0 (SD 7.54) kg/m^2^. Mean weight-change between baseline and endpoint (12 weeks, 10 group sessions) was −4.4 (SD 3.38) kg, and degree of weight change was associated with session attendance (F (9, 131) = 6.356, *p* < 0.0005). There were positive effects on participants’ weight, health-related behaviours, and quality of life. The intervention content (including the focus of nutritional recommendations) and structure were adapted during implementation to better suit national guidelines and local population needs. RE-AIM was found to be a useful framework for evaluating and adapting an existing evidence-based weight management programme in line with local population needs. This could be a more cost-effective approach, compared to developing new programmes, for delivering public health goals relating to obesity, nutrition, and physical activity.

## 1. Introduction

Obesity is widespread and costly economically and for health; 63% of adults in England were living with overweight (body mass index, BMI > 25 kg/m^2^) or obesity (BMI > 30 kg/m^2^) in 2018 [1]. Obesity increases the risk of type 2 diabetes, some cancers, and cardiovascular disease (CVD), shortening both total life expectancy and healthy life expectancy [2,3]. Obesity is also strongly associated with depression; this relationship is bi-directional and appears to be stronger in women than men [4,5]. Obesity is not only a health issue but also a social one; there is inverse socio-economic disparity in obesity distribution, which is especially pronounced amongst women [6]. The high prevalence of obesity means that it must be addressed primarily at a public health level through both prevention and treatment. Public health policy typically targets individuals either through weight-loss advice (for example, during a General Practitioner appointment), weight-management programmes (WMPs), weight-loss medication, or surgery. Which of these is indicated depends on an individual’s BMI status and existing co-morbidities [7]. The majority of weight-management support is provided in the community; one study found that of 91,000 individuals living with overweight and obesity in the United Kingdom (UK), 30% were offered advice, 20% referred to a WMP, and 14% given weight-loss medication [8], although an earlier UK study found lower rates: 20%, 4%, and 2%, respectively [9].

National Health Service (NHS) guidelines indicate that the recommended treatment for overweight and obesity is a structured, multicomponent lifestyle intervention addressing eating and physical activity through behavioural changes [10]. A Cochrane systematic review of behavioural interventions found greater weight loss in interventions combining elements of diet and exercise compared to interventions focused on either diet or exercise alone [11]. These interventions can also support maintenance of weight-loss, although maintenance of weight loss is notoriously difficult [12,13]. One potential reason for failure to maintain weight loss is that people are not equipped with the psychological strategies needed to sustain individual changes to diet and exercise behaviours. A systematic review found that effective interventions for weight loss used elements of behaviour change theory, including problem solving, self-monitoring, and other self-regulation strategies [14]. Another review of 122 studies found that interventions including self-regulation strategies (specifically self-monitoring combined with another self-regulation strategy) were more effective than interventions without these techniques [15].

Shape-Up is one such evidence-based behavioural weight-management intervention developed by the charity Weight Concern; it comprises a self-help manual for participants and a Group Facilitator’s Manual [16,17]. It has been widely disseminated; more than 20,000 people have participated in the programme via Public Health or NHS (based on numbers attending facilitator training, manuals sold, and service-evaluation data from service providers; information provided by Weight Concern). Shape-Up uses key behaviour-change techniques (including self-regulation strategies), such as graded goal-setting, self-rewards, self-monitoring, and relapse prevention. In contrast to many behavioural weight-management programmes, it is manualised so that it can be delivered by non-specialist facilitators, reducing overheads and allowing it to be delivered in community settings. Shape-Up is structured as an eight-week, manual-based group programme with an embedded peer-learning element to enhance engagement and self-efficacy. Participants are encouraged to volunteer to facilitate a peer-led section of each week’s topic, such as coping with triggers to overeating and food labels. With guidance and support from the facilitator, they then deliver this to their peers. Group members set their own goals relating to the session’s theme and are encouraged to focus on behavioural (relating primarily to nutrition and physical activity) and not weight-loss goals. Supporting this, training included taking a weight-neutral approach to interactions with participants. Organisations are charged a minimal fee for facilitator training and then need to buy participant manuals but are free to run as many groups as they like without any licensing or other costs.

Shape-Up was implemented in Rotherham (in the North of England) by the Places for People leisure provider (PfP) for three years between 2015–2018, reaching approximately 2000 people. Rotherham has one of the highest proportions of overweight and obesity, between 67–76% relative to the U.K. overall of 64% [18]. It also has high levels of deprivation [19], which is related to health issues, including obesity [20]. In 2015, Rotherham Metropolitan Borough Council launched the Weigh-Up initiative, acting as an umbrella for weight-management services across the borough and driving the NHS Rotherham’s Healthy Weight Commissioning Framework implementation. They commissioned PfP to deliver Shape-Up as a 12-week programme (and 12-week gym access). This intervention will be referred to throughout this paper as ‘PfP Shape-Up’ to distinguish it from the original Shape-Up programme.

Shape-Up has not been formally evaluated in an RCT; however, an early version was favourably evaluated in terms of acceptability, physical, and psychological outcomes over one-year follow-up [21]. This does not reflect the version that has subsequently been implemented in real-world settings. Whilst Shape-Up has been subject to service evaluations, none have been published, but typically public health providers have reported that participants achieve average weight losses of 2.8 kg (5.3% of original weight) over eight weeks (information provided by Weight Concern). Modified versions have been developed and evaluated, e.g., a version for individuals with intellectual disabilities [22] and a version for women with endometrial cancer [23]. Preliminary work for the latter involved an intervention mapping study that systematically described the intervention, identifying links between theory, behaviour-change strategies, and practice; key elements of the programme were identified as self-monitoring, behavioural goal setting, self-incentives, social support, and problem solving [24].

The RE-AIM framework [25,26,27] has been widely used to examine interventions pragmatically [28]; it can be used for planning and evaluation, including retrospectively. It focuses on external validity and sustainable programme adoption and includes the components of interventions that support effective implementation. An intervention’s success is conceptualised in terms of its reach (the proportion of the target population who participate), effectiveness (real world impact rather than in ideal conditions-efficacy), adoption (the number of providers who implement the intervention), implementation (how accurately the intervention is replicated and its realistic cost), and maintenance (how sustainable the effects of the intervention are). RE-AIM has been used in numerous contexts, for example, it was recently used to frame a qualitative study embedded in a trial to understand the translational potential of an intervention for treating comorbid obesity and depression in primary care [29]. In this study, participants and stakeholders were interviewed, and several elements of the RE-AIM framework were identified with potential for improving implementation. It was also used to guide the reporting in a recently published systematic review of weight-gain prevention interventions in young adults; the reporting of RE-AIM factors relating to external validity and generalizability were found to be limited, supporting the use of standardised reporting [30].

Given its wide usage and the novelty of the Shape-Up approach compared other community-based weight-management programmes, it is informative to understand which aspects of the Shape-Up intervention are effective and easy to implement and deliver. This evaluation examined implementation of the PfP Shape-Up using the RE-AIM framework. It examined secondary data on participants’ weight, behaviours, and psychological measures and how the intervention was implemented. The aim of this study was to examine the impact of Shape-Up as a public health intervention for weight management, delivered by PfP in Rotherham, and identify how the intervention was implemented in this real-life setting. We also aimed to explore factors influencing the effectiveness of the PfP implementation of Shape-Up as a public health intervention to reduce obesity in Rotherham.

## 2. Materials and Methods

### 2.1. Design

For practical reasons, this service evaluation used data for the last three-month wave (January–March 2018) of a three-year programme (April 2015–March 2018). Given that this evaluation was of delivery in a commissioned, non-clinical, real-word setting, a control condition was not practicable. It was undertaken in collaboration with PfP, who delivered the intervention and collected data as part of usual service delivery.

### 2.2. Intervention Content

Funding of PfP Shape-Up was dependent on delivery and outcomes meeting NICE guidelines, requiring the programme to be extended from eight weeks to 12 weeks [10]. Free access to the gym for 12 weeks or to specialist classes (for those meeting Tier 3 criteria) was included as part of the programme and was conditional on attending PfP Shape-Up sessions [31]. Participants received a gym induction and a copy of the Shape-Up handbook. There was no charge for participation or materials. Classes were offered in the evening, the daytime or separate daytime sessions for those meeting Tier 3 criteria.

Most sessions, except for first and last, which included data collection, followed a standard pattern. The fifth teaching session was designed to gather feedback and review progress so that course leaders could tailor provision to participants’ needs. All sessions were designed to introduce participants gradually to various weight-management behaviours. Weight goals were set at the first and last sessions and behavioural goals at the end of each session. Individuals were weighed in a private space upon arrival as part of the intervention to track progress. Those who did not lose or who gained weight were identified as potentially benefitting from additional support. Group sessions started with a discussion of progress made, obstacles encountered, and tips. The leader encouraged reflection and problem solving. The bulk of the meeting was a leader-led presentation and discussion based on that week’s topic. Individuals completed a written task in their manual and participated in group discussions. The end of each session was a round-up of the session (for example, food-diary or goal-setting reminders) and goal setting for the next week. Task sheets were provided for homework and any questions addressed. After the session, additional motivational slots based on motivational interviewing were offered to those identified at the start of the session as requiring additional support. During these, leaders might help the individual consider the impact of their diarised food intake or discuss strategies to reduce overeating.

### 2.3. Re-Aim Framework

The implementation of the Shape-Up programme in a real-world setting (PfP Shape-Up) was assessed using the RE-AIM framework examining reach, effectiveness, adoption, implementation, and maintenance of the delivered programme. RE-AIM is one of the most frequently used implementation-assessment frameworks [26,27,32]. It can be used retrospectively or at planning stages to assess external validity of an intervention, and it allows for quantitative and qualitative assessments [26]. It is designed for pragmatic use such that not every aspect of the framework needs to be assessed. Four of the five aspects of RE-AIM were relevant for the current evaluation (reach, effectiveness, implementation, maintenance), and these were used as a framework (see Appendix A).

The key areas for focus in this service evaluation were:

Reach: How participants were recruited, percentage of those initially expressing interest completed the course, representativeness of participants;

Effectiveness: Mean weight loss and other outcomes (behavioural and wellbeing measures);

Implementation: Adaptations that were made to the intervention. The framework element with the most indices is implementation, which allows assessment of how Shape-Up was adapted—adaptations being common consequences of implementation [33]; and

Maintenance: Elements of maintenance were discussed, although no data were available.

### 2.4. Data Collection

Quantitative data were collected by PfP course leaders on behalf of Rotherham Metropolitan Borough Council, U.K. as part of the Weigh-Up initiative. Some of the data had been entered directly onto electronic records (for example, ethnicity and 6- and 12-month follow-up data) and some stored as hardcopy. Only information on hardcopy was available for this evaluation, as the electronic data had been archived. Data were manually entered onto UCL’s Data Safe Haven by AS whilst based at Rotherham Leisure Complex. Data were collected for 141 participants from the final wave at the main three of the four delivery centres (Rotherham Leisure Complex, Wathen Market Surgery, and Maltby Leisure Centre); data were not able to be collected from the fourth centre due to capacity constraints. Data were entered for all participants starting the programme at these three centres for whom paperwork was available. Information was gathered from paperwork as well as brief, informal discussions with two course leaders, one of whom had overall responsibility as programme manager. They were able to provide clarifications around the course methodology and the content of a typical session and any challenges arising.

### 2.5. Measures

Data collected at baseline (first session attended) and endpoint (last planned session) were entered into the NHS Data Collection and Reporting Service (DCRS) system by PfP. Measures included those in the Shape-Up materials (“Just Starting” questionnaire, see Appendix A) as well as those added by PfP. Measures were administered by the course leader. These included:

**Weight (kg).** Taken at baseline and endpoint on the same set of scales. Participants wore light clothing and removed shoes. Weight from the last session attended by the participant was used as their endpoint weight if they did not attend their final planned session;

**Height (cm).** Measured at baseline;

**BMI.** Calculated from weight and height at baseline and endpoint. A cut-off of ≥25 was used to denote living with overweight and ≥30 for obesity, as per WHO guidelines [34];

**Waist circumference (cm).** Taken at baseline and endpoint using the same tape measure (which was standard but non-calibrated) each week for each group; and

**Self-completed questionnaires** were used to collect information at baseline and at the end of the intervention (therefore, endpoint measures were not available for those not attending their last intended session) on:

Age, sex, employment data, and smoking status (baseline only);

Daily fruit/vegetable consumption (from 0 to ≥6), 30-min bouts of moderate-intensity physical activity, and 20-min bouts of vigorous exercise weekly (0 to ≥5). These were compared to guidelines [35,36]. The fruit/vegetable question was a composite based on two biomarker-validated dietary questions [37]. The physical activity question was part of the standard Shape-Up questionnaire, and the exercise question was added by PfP; both were created for use with Shape-Up (see Appendix A, for questions).

Quality of life (QoL) was measured using the EQ-5D, a validated and widely used questionnaire [38]. It has two components: the EQ-5D-3L and the EQ-VAS (visual analogue scale). The EQ-5D-3L is a self-perceived categorical QoL score assessed on a three-point scale (1 being no limitations, 2 some, and 3 severe) along five dimensions (mobility, self-care, able to undertake usual activities, pain/discomfort, and anxiety/depression). Higher scores indicate more difficulty along each dimension, but these are not usually summated. EQ-VAS is a self-perceived health scale, ranked from 0 (worst) to 100 (best). There are no cut-offs for either measure.

Additional psychological measures developed for use with Shape-Up and included in the endpoint questionnaire were: ten measures of self-efficacy (questions with a five-point Likert scale of agreement to disagreement relating to dietary behaviours, behavioural strategies, control, motivation, and satisfaction with current weight) and 11 measures of satisfaction with the intervention and facilities. These had been tested as part of the “Just Starting” and Just Finished” Shape-Up questionnaires but were used solely at the endpoint in PfP Shape-Up (See Appendix A).

Costs for delivering the programme were obtained from the service providers through personal communication; a range was provided due to commercial sensitivities.

### 2.6. Statistical Analyses

Reach was assessed using descriptive analyses to identify baseline characteristics of participants. Data on the population of Rotherham were used as a comparator, e.g., mean age of a Rotherham resident was 39.6 years, and the last-recorded employment rate was 57% [39]. Effectiveness was judged through examining changes between baseline and last measured weight or endpoint questionnaire. Paired *t*-tests were used to assess change in weight (kg), waist circumference (cm), daily fruit and vegetable consumption, weekly moderate and weekly vigorous exercise, and EQ-VAS. “Robustness across sub-groups” (as per RE-AIM evaluation criteria) was tested by entry multiple regression model after testing data met assumptions. This tested the predictive value of age, sex, baseline BMI, employment status, self-reported readiness to change, and attendance on the course upon percentage of baseline weight lost and percentage of baseline waist circumference lost (dependent variables). EQ-5D-3L data were reported descriptively (modal change values were additionally calculated, see Appendix A). SPSS (IBM, Portsmouth, UK) was used for all analyses (SPSS 25, IBM). Implementation was assessed through direct comparison of Shape-Up (using the Shape-Up manual [17]) and PfP Shape-Up in terms of session structure, information delivered, role of the leader, key performance indicators (KPIs) measured, and assessment questionnaire used.

## 3. Results

Results are presented as per the RE-AIM framework.

### 3.1. Reach

Of the 377 individuals who expressed interest in or who were referred to the final wave of PfP Shape-Up (January–March 2018), 157 (42%) chose to participate (see Appendix A). Data were from the 141 participants who attended three of the four centres delivering the intervention (Rotherham Leisure Complex, Maltby Leisure Centre, Wathen Market Surgery). Information was not available about those not opting into the programme.

#### Characteristics of Participants at Baseline Compared with Non-Participants/Local Sample

Baseline demographic and anthropometric data are shown in Table 1; behavioural and quality of life data are shown in Table 2. Mean age of the sample was 48.9 (SD 14.47) years, older than the Rotherham average of 39.6 years. The sample was 67% female, and employment was lower than Rotherham’s last-recorded employment rate (41% vs. 57% respectively). Mean BMI was 38.0 (SD 7.54), with 14% living with overweight and 86% with obesity. Mean daily intake of fruit and vegetables was 3.6 (SD 1.56), mean frequency of 30-min bouts of moderate-intensity physical activity/week was 1.8 (SD 1.57), and mean frequency of 20-min bouts vigorous exercise/week was 1.0 (SD 1.41). Baseline QoL measures showed that, on the EQ-VAS Health Scale, the mean score was 37.0/100 (SD 28.71).

### 3.2. Effectiveness

Endpoint data and mean change from baseline are shown in Table 2.

#### 3.2.1. Measure of Primary Outcome

Mean weight change between baseline and endpoint was −4.4 (SD 3.38) kg, a significant reduction (*p* < 0.0005).

#### 3.2.2. Measure of Broader Outcomes/Multiple Criteria

A total of 83.0% of participants lost more than 1 kg and 73.8% more than 2 kg. Overall, 22.7% lost 3–5% of baseline weight, 40.5% more than 5%, and 36.9% lost less than 3% or gained weight. Waist measurement significantly reduced from baseline (118.6 (SD 17.53) cm) to endpoint (110.6 (SD 17.53) cm) (*p* < 0.0005). There were significant changes in the three behavioural measures. Mean endpoint fruit and vegetable intake was 4.9 (SD 1.42), a significant increase from baseline (*p* < 0.0005), on average, meeting national guidelines. Mean weekly number of 30-min bouts of moderate-intensity physical activity was 3.0 (SD 1.41) at endpoint and 2.5 (SD 1.55) for 20-min bouts vigorous exercise, both significant increases (*p* < 0.0005) and closer to government guidelines than baseline. For the QoL measures, self-rated health using the EQ-VAS improved significantly between baseline and endpoint (*p* < 0.0005). There appeared to be little change in QoL measured using the EQ-5D-3L, with the majority of participants reporting no change for all five categories (see Supplementary File, Appendix A for modal change values).

#### 3.2.3. Measure of Robustness across Subgroups

As shown in Table 3, the number of sessions attended predicted 28% of the variance in the percentage weight lost and 13% of the variance for waist-circumference reduction (*p* < 0.0005). Age, sex, baseline BMI, employment, and self-reported readiness to change were not significant predictors in either analysis.

#### 3.2.4. Measure of Short-Term Attrition (%) and Differential Rates by Patient Characteristics or Treatment Group

The mean number of sessions attended was approximately nine. Data were recorded on an intent to treat basis, including all those with available paper records. Snow caused the cancellation of nearly every session during the fifth week (fourth teaching session); this appeared to have affected subsequent attendance and reduced to ten the maximum number of sessions that it was possible for most participants to attend.

#### 3.2.5. Use of Qualitative Methods/Data to Understand Outcomes

The fifth teaching session was a review of progress, giving course leaders informal insight into potential reasons for individuals’ rate of progress. This was not formally documented but was able to be built into the remaining sessions so that they were tailored to participants’ requirements.

### 3.3. Implementation

#### 3.3.1. Adaptations Made to Intervention during Study (Not Fidelity)

The changes made to the original Shape-Up included the structure of the sessions, information delivered, leader’s role, the Key Performance Indicators (KPIs) measured, and assessment questionnaires used. There were ten teaching sessions rather than eight and an additional introductory motivational session. Session themes varied from Shape-Up, although changes primarily related to the order in which topics were delivered (Table 4). Sessions were delivered by course leaders rather than facilitated; the facilitator role recommended by Shape-Up was replaced by a PfP-employed sports or nutrition-focused instructor who presented the topic, drove discussions, corrected misunderstandings, and offered motivational support to participants. Detailed MS PowerPoint presentations were created by PfP. Weighing in and physical activity are recommended in PH53 Recommendation 9, and both were included in PfP Shape-Up [40]. Additional motivational sessions were designed to increase engagement by participants, showing them success stories and encouraging them to participate fully in the programme. Participants were offered free access to the gym for 12 weeks as an adjunct to the programme.

In terms of the intervention content, the information provided in the session topics tended to be more detailed than Shape-Up, as PfP participants were not expected to research the topics themselves. Some of the dietary advice differed from Shape-Up, for example, the more detailed discussion of glycaemic load rather than focusing on a balanced diet. The mid-way review session was used to explore specific topics of interest to the group, and topic themes in remaining sessions were shaped by this. The KPIs and the questionnaire and data portal that collected the information were aligned to what is required by PH53 [40]. They included additional self-assessed measures of quality of life.

#### 3.3.2. Cost of Intervention-Money

Programme cost was estimated at £300–£500/head, which, according to economic modelling undertaken by NICE (based on the potential health and economic consequences of intervening), would be cost effective if participants maintained a weight loss of 2 kg or more [40]. This loss was achieved by 73.8% of participants, but follow-up data on weight-maintenance were not available.

### 3.4. Maintenance (Individual Level)

Measurement of primary outcome (±comparison with a public health goal) ≥ 6 mo follow-up after final treatment contact and measure of broader outcomes:

Maintenance was assessed by PfP inviting all participants to attend in-person or telephone follow-up sessions. Individuals were assessed for weight loss/stasis/gain, fruit and vegetable consumption, and bouts of moderate and vigorous exercise at six and 12 months (entered into DCRS). These data were not available for the current service evaluation. There was also an informal discussion of their progress and their future aims, although this was not recorded. Self-efficacy measures were collected at endpoint only, so these cannot give insight into changes associated with the programme. These are shown in Table 5. It is notable that approximately half of the original sample responded. All respondents indicated generally high levels of self-efficacy for weight-management behaviours, with the highest ratings for ability to understand food labels and for confidence and motivation in making lifestyle changes. Satisfaction ratings for the programme were generally high, with the highest ratings relating to recommending the programme and on leaders’ skills and motivation.

## 4. Discussion

PfP Shape-Up had a positive effect on weight, with a significant and dose-dependent reduction over the course of the intervention. Other outcomes (waist measurement, fruit and vegetable consumption, moderate and vigorous exercise frequency, and quality of life measured with the self-rated EQ-VAS health scale) significantly improved, although there was no change for the EQ-5D-3L QoL measure. However, response rates for these endpoint measures were low, with just over half of the original intake and less than three quarters of those attending the final session completing these measures. The researcher (A.S.) observed that the psychological questions were difficult to see in the questionnaire, as they had had to be added to the original Shape-Up endpoint questionnaire; this could have affected completion rates. The use of the RE-AIM framework for this service assessment indicated that PfP implementation of Shape-Up deviated considerably from the original version of Shape-Up, although most of the core elements of the programme remained. Discussion below will encompass the various aspects of RE-AIM examined in more detail.

### 4.1. Reach

Participants’ employment rate was lower than the Rotherham and UK average; this may be due to obesity being associated with higher unemployment [41] or may simply reflect the fact that over half the participants attended classes during working hours. That Shape-Up appeared accessible and to have positive outcomes in this group despite indicators that participants were of low socio-economic status (SES) is encouraging, as evidence in lower SES groups is lacking. For example, a systematic review of the effectiveness of individual, community, and societal-level interventions at reducing socio-economic inequalities in obesity among adults found some evidence that community-based programmes can be effective among deprived groups, at least in the short term [42]. However, evidence of long-term effectiveness was limited, and few studies were conducted outside of the United States; only one study out of 12 was UK-based, and this intervention focused on dietary change with little behavioural support. Discussion with a course leader indicated those without fluent English may have found it difficult to access the course fully, potentially limiting equality of access or outcome. Qualitative research indicates that having a WMP course leader who speaks participants’ primary language is perceived positively [43]. It may be that future versions of Shape-Up could be adapted for English as a Second Language communities. The intervention primarily recruited via self-referral. It has been found that self-referral for weight-loss programmes attracts fewer men and younger people, broadly consistent with our findings [44]. The reliance on self-referral may also have resulted in uptake of more motivated participants, creating a potential response bias. A review found that referral could, however, pose a barrier to participation because of stigma, gatekeeping, and delay, so the possibility of self-referral may have had a positive impact in itself [45].

### 4.2. Effectiveness

Data from this evaluation indicated that 22.7% lost 3–5%, 40.5% more than 5%, and 36.9% lost less than 3% of baseline bodyweight (the latter including some who gained weight). This meets the then Department of Health’s best practice guidance for weight management services [46]. A NICE economic modelling report found the heavier and older the person, the more likely an intervention is to be cost-effective [41]. This study, however, found no impact of baseline weight nor age on the percentage of weight lost. Results were suggestive of there being some improvements in self-reported health-related behaviours and quality of life. The results from service evaluations of commercial weight-management programmes in large samples of UK adults, who were referred by the NHS, found similar reductions in weight and BMI as the current study [47,48].

### 4.3. Implementation

Delivery of the programme was in weekly groups as per the original Shape-Up. Group delivery is typically preferred by WMP participants [12,49]. The groups potentially offered a source of support, which has been positively associated with WMP success [45]. Furthermore, positive group dynamics have been positively associated with weight loss [50]. Some work has also found that those who are more socially isolated benefit more from group attendance [51]. Frequent meetings have been found to be associated with greater weight loss [12,52]. The use of social media, such as the ongoing Facebook support group, may further have promoted this. Some changes were made to the programme; the key differences between the original programme and that delivered for the current evaluation were: the removal of the pure peer-support-led approach; expert-led rather than expert-facilitated; addition of sessions, including motivational sessions; supplementation of healthy balanced diet guidance with glycaemic load information; and addition of gym membership over the 12 week programme. The peer-led elements of Shape-Up are a core part of the programme, and removing this may have an impact on self-efficacy, which could possibly impact maintenance of behaviour changes. Alternatively, participants might have been intimidated by being required to give a presentation, or lower quality of presentations or content may have impacted results. Peer-led approaches have not been widely used in adult weight-management interventions, although more commonly for diabetes prevention and management (e.g., [53,54]). One example of a peer-led weight-management intervention delivered in the United States had promising weight-loss outcomes over a period of seven years for participants remaining engaged with the programme [55]. The authors of this study attributed this success to participants engaging long-term in the programme; the peer-led approach may have facilitated this. Another study that evaluated the impact of a peer-led workplace weight-management intervention, based on the NHS Choices weight-loss guide, found significant weight reductions over 12 weeks, although it is not clear if the outcomes were due to the peer-led aspect or the workplace setting [56]. Shape-Up’s peer-led approach embedded in a structured behavioural programme is novel and has potential to enhance outcomes through greater engagement of participants, as trust and influence may be greater with a peer leader compared to a health professional as well as keeping delivery costs to a minimum. Delivery of complex behaviour-change strategies within this context has potential to be particularly powerful and is worthy of further study.

The role of group dynamics within weight management has not been frequently studied; one study found that lower group conflict was associated with greater weight loss, and a relationship mediated by session attendance and engagement with self-monitoring and wanting to be accepted by the group was associated with higher attendance [50]. This suggests that skilful group facilitation could enhance outcomes and vice versa, meaning that potentially, a move away from facilitation in the current programme delivery could have negatively impacted outcomes, but we have no direct evidence for this. Context is an important consideration in implementation evaluation, and adaptations made by PfP may have improved uptake or outcomes for their target population but might not do so under other circumstances [32]. Some of the PfP adaptations that have been found to positively influence weight loss are the inclusion of exercise and supportive relationships with providers and peers [45] and motivational sessions [57]. Review authors suggested the underlying mechanisms may be that relationships prompted accountability and bonding, and healthy behaviours improved self-efficacy; this may also be the case for the motivational sessions. QoL measures are seen as useful secondary outcomes and deemed essential in weight-management programme delivery [31,40]. The impact of the changes to the dietary components are unclear, but studies have found similar weight losses in groups following different diets, so changes to the nutritional components of the programme may not have had a substantial impact (e.g., [58]). Discussion with a course leader raised the issue of whether the balanced diet approach (primarily a low-fat, higher-carbohydrate diet) was still relevant. This suggests greater enthusiasm for the modified dietary guidance by group leaders.

### 4.4. Maintenance

Assessment was difficult in terms of actual follow-up results because data from the NHS DCRS database were not accessible. Procedurally, the Programme Manager indicated all participants were invited for an in-person or over the telephone follow-up at both six and 12 months. This did not occur for this final wave of participants because funding ended. There was no PfP analysis of demographic patterns of maintenance of weight loss nor of attrition before and during the three-year programme. It is possible that the voluntary nature of follow-ups risked bias in response, as those experiencing greater success may be more likely to remain in contact. An understanding of the persistence of outcomes is important in evaluations of lifestyle weight-management programmes [13].

There are limitations to the work. The sample size included in this analysis was limited, and the lack of access to the NHS DCRS portal meant that data at follow-up and baseline on ethnicity and SES were not available. Endpoint data were only available for participants who attended the final session, resulting in missing data and potential bias. We were only able to collect data from three of the four centres included in the final wave of Shape-Up due to capacity constraints; we do not anticipate that this would have introduced bias but cannot rule this out. The lack of a control group means that only preliminary conclusions can be made about efficacy. Since data were collected by service providers, there is potential for bias, and the self-report nature of the questionnaire measures puts them at risk of social desirability. Some of the limitations highlight the challenges from this being a real-world evaluation rather than a controlled trial; we had limited information about some of the measurement procedures, for example, the brand of weighing scales and height measurer and the methods used to measure height and waist circumference. We had only limited information about the cost of delivering the programme; we did not have information about how costs were calculated and were only provided with a range. This study used RE-AIM retrospectively to evaluate the implementation of Shape-Up, but future work incorporating the RE-AIM framework at planning stages would offer greater insights into the extent of reach possible, the drivers of efficacy and uptake amongst providers, the adaptations required, and the ongoing implementation of the intervention. Sex disaggregation of outcomes and demographics from larger samples would also add insight. Assessing the mechanisms through which PfP Shape-Up brought about change and whether this was enhanced by the adaptations identified was not possible—this lack of transparency has been characterised as a “black box” [59,60]. However, the use of real-world data and consideration of findings in the context of a sound theoretical framework (RE-AIM) gives external validity to the work. RE-AIM looks beyond the effectiveness of an intervention to who is able to participate, what proportion of relevant organisations choose to deliver the intervention, how it has been implemented, and how long delivery by providers and effects in participants are maintained. This gives a broader understanding of why an intervention may or may not work in a real-world setting [61].

### 4.5. Implications for Future Work

Shape-Up has been widely used by commissioning bodies and their service providers who are obliged to follow NICE guidelines and will receive a “Quality Premium” if they meet a set of targets [10]. There would, therefore, be value in looking to align Shape-Up with those guidelines. The content of Shape-Up may need to change as the evidence base and the policy guidelines change. There may be value in a more agile Shape-Up that could adapt and change elements as needed by commissioning requirements or changes in dietary and other guidelines. This would be consistent with Public Health England’s new guidance of tailoring obesity programmes to context [62]. There may also be value in offering the participant questionnaires in an adaptable format so that KPIs could be easily added or removed, thus promoting ease of use and response.

Shape-Up is an evidence-based programme, and an obvious improvement would be to incorporate evidence from service evaluations, creating a virtuous cycle of evidence augmentation and implementation. This information could also be fed back to the commissioners and providers; a commissioner from Rotherham is quoted in a case-study as saying “We’ve struggled to engage academic colleagues to evaluate… I..[would like to] find out if what we’re doing is effective or not in comparison to what others are doing” [45]. The scope of this service evaluation did not include the acceptability of the Shape-Up intervention to those delivering it nor the rates of adoption across different providers. This is, however, an important element in an intervention’s impact because if it is challenging to provide, then it is less likely to be offered to service users [63]. Future work could look at what prompts local authorities to use Shape-Up. In addition, future service evaluations could use a checklist to note which approach and which components had been used, allowing greater insight into which of those had greater effect, reducing the “black box” effect and permitting greater replicability.

## 5. Conclusions

This service evaluation is novel in using RE-AIM to perform a structured detailed service evaluation of the real-world use of the Shape-Up intervention to support progress towards public health goals. In assessing the intervention along different dimensions, it was possible to see that a wider programme of standardised service evaluations would offer valuable evidence towards further refining and improving this evidence-based intervention. Structured weight-management programmes, such as Shape-Up, take considerable amounts of expertise, time, and financial resource to develop. However, tailoring weight-management interventions to meet the needs of local populations is also required if such programmes are to deliver public health outcomes. It makes economic sense for existing evidence-based approaches to be adapted using a structured approach that supports research and evaluation efforts to understand what works for whom. This study suggests that RE-AIM would be a useful framework for evaluating and adapting evidence-based weight-management programmes.

## Figures and Tables

**Table 1 nutrients-13-02807-t001:** Demographic, anthropometric, and readiness to change characteristics of sample at baseline.

Baseline Characteristics of Sample	*N*	Mean (SD) at Baseline or %
**Sex**	**141**	
Female	95	67%
Male	46	33%
**Age**	**139**	48.9 (14.47) years
**Employment**	**135**	
Employed	55	41%
Unemployed	25	19%
Student	3	2%
Long-term sickness	13	9%
Retired	31	23%
Other	8	6%
**Smoker**	**134**	
Non-smoker	120	90%
Smoker	14	10%
**Weight (kg)**	**141**	105.8 (23.90)
**Height (cm)**	**139**	166.8 (9.05)
**Body Mass Index (BMI)**	**139**	38.0 (7.54)
**BMI category**	**139**	
Living with overweight	20	14%
Living with obesity	119	86%
**Waist measurement (cm)**	**140**	118.6 (17.53)
**Readiness to change (on scale of 1–10, 10 most ready to change)**	**134**	9.1 (1.37)

(bold indicates headers).

**Table 2 nutrients-13-02807-t002:** Anthropometric and behavioural and psychological measures: baseline, endpoint, and change values.

	*N* at Baseline	Mean (SD) at Baseline or %	*N* at Endpoint	Mean (SD) at Endpoint or %	Mean (SD) Change from Baseline−Endpoint	Significance in Difference
Weight (kg)	141	105.8 (23.90)	141	101.1 kg (22.76)	−4.4 (3.38)	*t* = (140) 14.582, *p* < 0.0005
Body Mass Index (BMI) (kg/m^2^)	139	38.0 (7.54)	137	36.2 (7.36)	−1.7 (1.32)	*t* = (136) 15.003, *p* < 0.0005
Waist Measurement (cm)	140	118.6 (17.53)	115	110.6 cm (19.70)	−6.1 (3.85)	*t* = (114) 16.315, *p* < 0.0005
Fruit And Vegetable Portions/Day	135	3.6 (1.56)	77	4.9 (1.42)	+1.4 (1.70)	*t* = (76) −7.099, *p* < 0.0005
Number of 30-min Bouts of Moderate-Intensity Activity/Week	134	1.8 (1.57)	77	3.0 (1.41)	+1.2 (1.60)	*t* = (76) −6.683, *p* < 0.0005
Number of 20-min Bouts of Vigorous Exercise/Week	132	1.0 (1.41)	78	2.5 (1.55)	+1.5 (1.64)	*t* = (77) −8.001, *p* < 0.0005
EQ-VAS Scale ^^^ (Range 0–100)	133	37.0 (28.71)	60	66.5 (18.03)	+25.9 (28.71)	*t* = (59) −8.844, *p* < 0.0005
EQ-5D−3L *						
QoL Mobility	126					
1	89	71%	73	69.9%		
2	37	29%	22	30.1%		
3	0	0%	0	0%		
QoL Self-Care	127		73			
1	113	89%	65	89.0%		
2	12	9%	8	11%		
3	1	1%	1	0%		
QoL Usual Activities	127		73			
1	92	72%	52	71.4%		
2	34	27%	20	27.4%		
3	1	1%	1	1.4%		
QoL Pain/Discomfort	124		74			
1	62	50%	37	50.0%		
2	50	40%	30	40.5%		
3	12	10%	7	9.5%		
QoL Anxiety/Depression	128		74			
1	76	59%	48	64.9%		
2	45	35%	18	24.3		
3	7	6%	8	10.8%		

^^^ increase indicates perceived improvement. * Scoring: 1, no limitations; 2, some limitations; 3, severe limitations; decrease, improvement; increase, deterioration (see Appendix A for modal change values).

**Table 3 nutrients-13-02807-t003:** Regression analyses of predictors of weight and waist lost.

Independent Variable	Regression Co-Efficient (B)	95% CI	Standardised Regression Co-Efficient (β)	*t*	*p*
**Weight lost (Model Adj *R*^2^ = 0.280 (Std. error: 2.83%))**					
Baseline BMI	−0.026	(−0.046–0.098)	−0.059	0.719	0.473
Age	0.028	(−0.013–0.068)	0.123	1.367	0.175
Sex	−0.022	(−1.20–1.158)	−0.003	−0.037	0.971
Readiness to change	0.044	(−0.351–0.439)	0.018	0.222	0.825
Employed or not	−0.076	(−1.271–1.118)	−0.011	−0.127	0.899
Number of sessions attended	0.833	(0.589–1.078)	0.554	6.757	<0.001
**Waist circumference lost (Model Adj *R*^2^ = 0.126 (Std. error: 3.58%))**					
Baseline BMI	−0.068	(−0.167–0.031)	−0.135	−1.368	0.175
Age	0.023	(−10.984–7.190)	0.087	0.786	0.434
Sex	−1.184	(−2.84–0.479)	−0.150	−1.416	0.160
Readiness to change	−0.119	(−0.700–0.462)	−0.042	−0.406	0.686
Employed or not	−0.227	(−1.883–1.428)	−0.029	−0.273	0.785
Number of sessions attended	1.205	(0.574–1.836)	0.378	3.797	<0.001

**Table 4 nutrients-13-02807-t004:** Comparison of the Shape-Up manual and how the programme was delivered in Rotherham in the period studied (January–March 2018).

Sessions	Shape-Up Manual	Rotherham 2018 * (Number in Brackets Indicate Original Shape-Up Programme Session Number)
Introduction		Motivational Session: Questionnaire and Measurements
1	Preparing to Shape-Up	Introduction: Prepare to Shape-Up (1)
2	Keeping to a regular eating pattern	Keeping to a regular eating pattern (2)
3	Physical activity	Eating a balanced diet (4)
4	Eating a balanced diet	Food serving sizes (5)
5	Food serving sizes—how to cut the quantity	What’s in my drink (Rotherham session)
6	External triggers	Mid-way review of progress and review of physical activity (Rotherham session) This entailed small group discussions asking people to share their experiences, provide feedback, and buddy up with others
7	Internal triggers	Food labels (some of 8), using British Heart Foundation tools
8	Food labels and the Shape-Up plan	Internal and external triggers (6 and 7). The Programme Manager indicated that this was a sensitive topic for many participants, and it was important to develop trust within the group before discussing this topic
9		Takeaways, eating out, and nutrition-related ageing (this linked into discussions from the mid-way review and a topical television programme many participants had watched
10		Final reviews: Shape-Up Change Plan (some of 8)
6 months follow-up	Follow-up questionnaire and meeting. Feedback to group and discussion of self-regulation and goal setting. Self-determined SMART goals. Handouts provided.	KPIs: Weight change: % of individuals who gained, lost, and/or maintained their weight Fruit and vegetable consumption, 30-min bouts exercise/week, 20-min bouts vigorous exercise/week.
12 months follow up	Follow-up questionnaire and meeting. Feedback to group and discussion of self-regulation and goal setting. Self-determined SMART goals. Handouts provided.	KPIs: Weight change: % of individuals who gained, lost, and/or maintained their weight Fruit and vegetable consumption, 30-min bouts exercise/week, 20-min bouts vigorous exercise/week.

* PfP Shape-Up topics are shared with the kind permission of PfP and remain their intellectual property; KPI, key performance indicator.

**Table 5 nutrients-13-02807-t005:** Self-Efficacy and Satisfaction Measures at Endpoint.

Statement	*N*	Mean (SD)
Self-efficacy questions		
Have a regular pattern of eating	80	4.0 (0.97)
I always have a balance of different types of food in my diet	78	3.9 (0.90)
I set effective lifestyle goals and work towards them	79	4.0 (0.82)
I am in control of my food portion sizes	77	4.1 (0.87)
I know what appropriate food portion sizes are for losing weight	80	4.3 (0.67)
I am able to manage triggers that lead to unhealthy behaviours (may include things like your mood, or the sight/smell of tempting food)	78	3.9 (0.83)
I am confident that I understand the information on food labels	79	4.4 (0.68)
I feel in control of my eating habits	76	4.1 (0.73)
I feel confident and motivated to make any lifestyle changes	81	4.4 (0.74)
I am happy with my current weight	79	3.0 (1.29)
Satisfaction questions		
Learnt new skills to help control weight	80	4.6 (0.74)
Found the group sessions enjoyable	79	4.7 (0.64)
Group sessions relevant to needs	80	4.5 (0.69)
The group leader(s) were skilled and effective	79	4.7 (0.75)
The group leader(s) were motivating	79	4.7 (0.73)
The Shape-Up programme (workbook and groups) lived up to expectations	80	4.4 (0.79)
The group sessions helped with weight management	80	4.5 (0.78)
Would recommend Shape-Up groups to a friend who wanted to manage their weight	79	4.7 (0.72)
The facilities were adequate and comfortable	79	4.4 (0.80)
The time allocated for the session was sufficient for the content delivered	80	4.4 (0.89)
I would be interested in finding out more about additional services and membership offers to help me maintain my new weight	80	4.5 (0.83)

Scoring: agreement on Likert 5 point scale, 1 = strongly disagree, 5 = strongly agree.

## Data Availability

The data for the study were routinely collected data and belong to PfP; they are therefore not available for sharing.

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
