# Peer review of "‘Shape-Up’, a Modified Cognitive-Behavioural Community Programme for Weight Management: Real-World Evaluation as an Approach for Delivering Public Health Goals"

_nutrients, 2021, doi:10.3390/nu13082807_

Round 1
Reviewer 1 Report
Shape-Up… is an extremely thorough and interesting evaluation of a real-world weight loss program. The RE-AIM evaluation presented will inform many future programs as they create true evidence-based programs. I only have a few comments.
In the abstract, line 26, I believe the BMI value need a number in the tenths place.
On page 5, 2.5 Measures, it would be helpful to have more detailed description of who assessed the various measures, and how the data were collected. I think this is very important because there was such a low response rate for the questionnaires at the follow-up. You discuss that they may have been lost in the shuffle of papers at the end. But why didn’t that happen during the baseline measures. This information will help inform future studies on best practices for more complete data capture.
Page 5, Line 214. Also for waist circumference, how was it measured? What is percentage waist circumference
Line 220. How were exercise / vigorous exercise assessed?
Page 6, Line 259. Since 220 people chose not to participate, it would be very informative to have some data as to why. That way future studies could try to develop ways to overcome those obstacles.
Page 6, Line 269. There appears to be no space between ‘with14%’
Tables 1 &2. For clarity, I think it would be more helpful to have Table I be just demographics (sex through smoker). Then Table II could present baseline and endpoint values for each of the outcome variables. I don’t think we need all of the details regarding the QoL subscales – the current layout is difficult to parse.
Author Response
Shape-Up… is an extremely thorough and interesting evaluation of a real-world weight loss program. The RE-AIM evaluation presented will inform many future programs as they create true evidence-based programs. I only have a few comments.
Thank you for your positive feedback.
In the abstract, line 26, I believe the BMI value need a number in the tenths place.
We have amended this.
Line 26: “Mean BMI was 38.0 (SD 7.54) kg/m2.”
On page 5, 2.5 Measures, it would be helpful to have more detailed description of who assessed the various measures, and how the data were collected. I think this is very important because there was such a low response rate for the questionnaires at the follow-up. You discuss that they may have been lost in the shuffle of papers at the end. But why didn’t that happen during the baseline measures. This information will help inform future studies on best practices for more complete data capture.
Thank you for pointing this out. We specified that the course leader took the weight measures, but should have specified that the course leader administered all of the measures. We have therefore amended this.
Line 228-229: “Measures were administered by the course leader.”
Page 5, Line 214. Also for waist circumference, how was it measured? What is percentage waist circumference
We included reference to percentage waist circumference by mistake so have removed this. We also removed reference to percentage weight change. (lines 238-242)
We have added that the tape measure was standard but non-calibrated. We do not have further information about the methods used. We have therefore added this to the limitations section (related to this, we have also added limitations re measuring weight and height).
Lines 238-29: Taken at baseline and endpoint using the same tape measure (which was standard but non-calibrated) each week for each group.
Lines 617-621: Some of the limitations highlight the challenges from this being a real-world evaluation rather than a controlled trial; we had limited information about some of the measurement procedures for example, the brand of weighing scales and height measurer and the methods used to measure height and waist circumference.
Line 220. How were exercise / vigorous exercise assessed?
We have added details of the assessment of vigorous exercise and additional clarity for the measurement of moderate intensity activity. We have added the wording for the vigorous activity question to the supplementary file.
Lines 247-249: “The physical activity question was part of the standard Shape-Up questionnaire and the exercise question was added by PfP, both were created for use with Shape-Up (see Supplementary File, Figure S1, for questions).”
Page 6, Line 259. Since 220 people chose not to participate, it would be very informative to have some data as to why. That way future studies could try to develop ways to overcome those obstacles.
We do not have information about the reasons for people not opting into the programme and have added this to the results section.
Lines 302-303: Information was not available about those not opting into the programme.
Page 6, Line 269. There appears to be no space between ‘with14%’
Thank you- we have amended this.
Tables 1 &2. For clarity, I think it would be more helpful to have Table I be just demographics (sex through smoker). Then Table II could present baseline and endpoint values for each of the outcome variables. I don’t think we need all of the details regarding the QoL subscales – the current layout is difficult to parse.
Thank you for this suggestion. We have moved the baseline behavioural and quality of life measures from Table 1 to Table 2. Anthropometric outcomes were kept in Table 1, but additionally included in Table 2 for consistency. We agree that the quality of life results are confusing, we have therefore moved the modal changes for quality of life to the supplementary file (Table S3). This has also been amended in the methods section.
Lines 289-290: EQ-5D-3L data were reported descriptively (modal change values were additionally calculated, see Supplementary File, Table S3).
Lines 306-307: Baseline demographic and anthropometric data are shown in Table 1, behavioural and quality of life data are shown in Table 2
Lines 345-347: There appeared to be little change in QoL measured using the EQ-5D-3L, with the majority of participants reporting no change for all five categories (see Supplementary File, Table S3 for modal change values).
Reviewer 2 Report
The study is interesting because many intervention projects are carried out in the field, but they are rarely published and evaluated. It is essential to know the effectiveness of intervention programs before proceeding to replicate them elsewhere.
- In the introduction, they should refer to the relationship between BMI and depression, especially in women. Cohort studies show an association between obesity, overweight, and depression, or between diet and depression. It is a two-way street, obesity increases the risk of depression, and depression increases the risk of obesity.
- The material and methods section indicates that the RE-AIM method is used for the evaluation.
- As this is not a very generalized evaluation system, please explain it in more detail so that unfamiliar readers can understand it without consulting the original article. In the bibliography, there is a reference to a review article from years ago. Would you please provide some recent examples of its use?
- If we exclude references to web pages, the obsolescence rate of the bibliography is seven years. The bibliography should be updated.
- The authors accessed the web references two years ago. Please check them, verify that the link is correct, and update the access date.
- In lines 194-195, the authors say, "Data were collected from 141 participants from the last wave (...) Data were entered for all participants starting the program at these three centers, for whom paperwork was available". It is stated that the data are from 3 of the four centers in which there were participants. Table s2 presents the entire recruitment process. But it is not clear whether table s2 refers only to the three centers' participants or those from all four centers.
We do not know how many participants were in this wave neither the proportion of people participating in the evaluation. The authors should specify participation. It is necessary to discuss the center not included and the reasons for its non-inclusion. In the discussion, it would be required to evaluate or rule out any possible bias.
- Would you please provide in the material and methods section the BMI intervals for overweight and obesity, presented in Table 1
- As an EFL (English as a Foreign Language) speaker, I don't understand Table 1. why table 1 says "living with obesity," "living with overweight."? what does "living with" mean? Is it a colloquial term or idiom in your region, a politically correct wording, or other technical implications? I am confused. I would be grateful if you could comment on this in your answers.
- In table 3, it is incorrect to put P= 0.0000 because this is 0. You should write p < 0.0001.
- It would be interesting if you could provide in the text the Pearson correlation (r), as well as the coefficient of determination (R2) of the association between the number of sessions attended with the reduction in waist circumference and weight reduction.
- Would you please explain how you calculated the program's cost per person and what inputs you included?
- If you had divided the program's cost by the number of people, you should have obtained a single per capita cost figure. I don't understand why you give a range of costs, how did you calculate it? Can you explain it?
- Although a bibliographic reference is provided, it would be interesting to explain why the intervention is cost-effective in more detail. In addition, it would be necessary to indicate from what point of view, from the point of view of society as a whole, health services, etc.
Author Response
The study is interesting because many intervention projects are carried out in the field, but they are rarely published and evaluated. It is essential to know the effectiveness of intervention programs before proceeding to replicate them elsewhere.
Thank you for this positive feedback.
- In the introduction, they should refer to the relationship between BMI and depression, especially in women. Cohort studies show an association between obesity, overweight, and depression, or between diet and depression. It is a two-way street, obesity increases the risk of depression, and depression increases the risk of obesity.
We have added reference to this in the introduction.
Lines 44-47: “Obesity is also strongly associated with depression, this relationship is bi-directional and appears to be stronger in women than men (Milaneschi et al, 2019; Baldini et al, 2021.”
- The material and methods section indicates that the RE-AIM method is used for the evaluation.
- As this is not a very generalized evaluation system, please explain it in more detail so that unfamiliar readers can understand it without consulting the original article. In the bibliography, there is a reference to a review article from years ago. Would you please provide some recent examples of its use?
Thank you for this suggestion. We have expanded the description of RE-AIM in the introduction and included recent examples of its use. We have added the web link for the RE-AIM website and removed some of the older references in the introduction and methods sections.
Lines 123-125: “It focuses on external validity and sustainable programme adoption and includes the components of interventions which support effective implementation.”
Lines 130-138: “RE-AIM has been used in numerous contexts, for example, it was recently used to frame a qualitative study embedded in a trial to understand the translational potential of an intervention for treating comorbid obesity and depression in primary care (Lewis et al, 2021). In this study, participants and stakeholders were interviewed and several elements of the RE-AIM framework were identified with potential for improving implementation. It was also used to guide the reporting in a recently published systematic review of weight gain prevention interventions in young adults; the reporting of RE-AIM factors relating to external validity and generalizability were found to be limited, supporting the use of standardised reporting (Haire-Joshu et al, 2021).”
- If we exclude references to web pages, the obsolescence rate of the bibliography is seven years. The bibliography should be updated.
We have updated older references where more recent ones are available, this includes the following additions:
WCRF/AICR, 2018; Avekwe et al, 2020; Wadden et al, 2020; LeBlanc et al, 2018; Tate et al, 2019
- The authors accessed the web references two years ago. Please check them, verify that the link is correct, and update the access date.
Thank you for this observation, we have updated this.
- In lines 194-195, the authors say, "Data were collected from 141 participants from the last wave (...) Data were entered for all participants starting the program at these three centers, for whom paperwork was available". It is stated that the data are from 3 of the four centers in which there were participants. Table s2 presents the entire recruitment process. But it is not clear whether table s2 refers only to the three centers' participants or those from all four centers.
We do not know how many participants were in this wave neither the proportion of people participating in the evaluation. The authors should specify participation. It is necessary to discuss the center not included and the reasons for its non-inclusion. In the discussion, it would be required to evaluate or rule out any possible bias.
Thank you for picking this up, we have clarified that data were collected from 3 of the 4 centres due to capacity constraints and also added clarification to Table S2 that it includes participants from the 4 centres in this wave.
Lines 216-219: “Data were collected for 141 participants from the final wave, at the main three of the four delivery centres (Rotherham Leisure Complex, Wathen Market Surgery and Maltby Leisure Centre); data were not able to collected from the fourth centre due to capacity constraints.”
We added reference to this in the limitations section.
Lines 612-614: We were only able to collect data from three of the four centres included in the final wave of Shape-Up due to capacity constraints, we do not anticipate that this would have introduced bias but cannot rule this out.
The following footnote was added to Table S2: (NB/ this table includes data from the whole of the wave, which includes participants from four leisure centres, note that only three of these centres were included in the current service evaluation).
- Would you please provide in the material and methods section the BMI intervals for overweight and obesity, presented in Table 1
We have added this to the methods.
Lines 235-237: BMI. Calculated from weight and height at baseline and endpoint. A cut-off of ≥25 was used to denote living with overweight and ≥30 for obesity, as per WHO guidelines [34].
- As an EFL (English as a Foreign Language) speaker, I don't understand Table 1. why table 1 says "living with obesity," "living with overweight."? what does "living with" mean? Is it a colloquial term or idiom in your region, a politically correct wording, or other technical implications? I am confused. I would be grateful if you could comment on this in your answers.
Thank you for this observation. This is the recommended terminology for referring to obesity with person-first language. Eg- World Obesity https://www.worldobesity.org/what-we-do/our-policy-priorities/weight-stigma
- In table 3, it is incorrect to put P= 0.0000 because this is 0. You should write p < 0.0001.
Thank you- we have amended this.
- It would be interesting if you could provide in the text the Pearson correlation (r), as well as the coefficient of determination (R2) of the association between the number of sessions attended with the reduction in waist circumference and weight reduction.
We appreciate this suggestion, but we do not readily have access to these data, therefore have opted to leave it as it is, we hope that this is acceptable.
- Would you please explain how you calculated the program's cost per person and what inputs you included?
Unfortunately, we were only provided with the costs, not the inputs, and we were only provided with a range. We have added reference to this in the methods and also mentioned this as a limitation.
Lines 275-276: “Costs for delivering the programme were obtained from the service providers through personal communication, a range was provided due to commercial sensitivities.”
Lines 621-623: “We had only limited information about the cost of delivering the programme; we did not have information about how costs were calculated and were only provided with a range.”
- If you had divided the program's cost by the number of people, you should have obtained a single per capita cost figure. I don't understand why you give a range of costs, how did you calculate it? Can you explain it?
As per above point.
- Although a bibliographic reference is provided, it would be interesting to explain why the intervention is cost-effective in more detail. In addition, it would be necessary to indicate from what point of view, from the point of view of society as a whole, health services, etc.
We have added clarifications on this, including what the comparison based their modelling on.
Lines 470-472: Programme cost was estimated at £300-£500/head, which according to economic modelling undertaken by NICE (based on the potential health and economic consequences of intervening), would be cost effective if participants maintained a weight loss of 2kg or more [40].